# A Cross-Sectional Study on the Associations between Physical Activity Level, Depression, and Anxiety in Smokers and Ex-Smokers

**DOI:** 10.3390/healthcare10081403

**Published:** 2022-07-27

**Authors:** Ángel Denche-Zamorano, David Manuel Mendoza-Muñoz, Raquel Pastor-Cisneros, José Carmelo Adsuar, Jorge Carlos-Vivas, Juan Manuel Franco-García, Jorge Pérez-Gómez, María Mendoza-Muñoz

**Affiliations:** 1Promoting a Healthy Society Research Group (PHeSO), Faculty of Sport Sciences, University of Extremadura,10003 Caceres, Spain; andeza04@alumnos.unex.es (Á.D.-Z.); jadssal@unex.es (J.C.A.); jorgecv@unex.es (J.C.-V.); 2Social Impact and Innovation in Health (InHEALTH), University of Extremadura, 10003 Caceres, Spain; raquelpc@unex.es; 3Health Economy Motricity and Education (HEME), Faculty of Sport Sciences, University of Extremadura, 10003 Caceres, Spain; jmfrancog@unex.es (J.M.F.-G.); jorgepg100@unex.es (J.P.-G.); 4Research Group on Physical and Health Literacy and Health-Related Quality of Life (PHYQOL), Faculty of Sport Sciences, University of Extremadura, 10003 Caceres, Spain; mamendozam@unex.es; 5Departamento de Desporto e Saúde, Escola de Saúde e Desenvolvimento Humano, Universidade de Évora, 7004-516 Évora, Portugal

**Keywords:** lifestyle, mental illness, risk factors, smoking

## Abstract

Introduction: depression and anxiety is one of the most relevant public health problems. The link between smoking and depression has been demonstrated. Regular physical activity (PA) could act as a protector against mental health diseases. Objectives: (1) to explore the prevalence of depression and anxiety in relation to sex and to the condition of smoker and ex-smoker, (2) to study the differences in prevalence proportions according to the frequency and level of PA, and (3) to calculate the probability of presenting depression or anxiety according to the frequency and level of PA. Methods: a cross-sectional study was conducted, based on data extracted from the 2014 and 2020 Spanish European Health Survey and the 2017 Spanish National Health Survey. The sample included 9524, 7813, and 9079 participants, respectively. Descriptive analysis was performed (comparisons using a Chi-square test and z-test for independent proportions). Probability odds ratios of anxiety and depression were calculated according to PA. Results: women had a higher prevalence of depression and anxiety than men (*p* < 0.001–0.003). Higher levels of prevalence were observed in inactive people versus very active or active people (*p* < 0.001). Inactive people had a higher risk of depression and anxiety compared to very active people. Conclusion: inactive smokers and ex-smokers had higher levels of prevalence of depression and anxiety than active and very active people. Physical inactivity could increase the risk of suffering these mental disorders in this population. This could affect women more than men.

## 1. Introduction

Mental health problems are considered to be one of the leading causes of disability. Moreover, this problem is a threat to public health in the world’s population due to the progression of the pathology, the increase in prevalence, and the difficulties in therapeutic control [1,2]. In particular, depression and anxiety are considered to be highly relevant indicators of mental health, and can have a harmful effect on people who do not receive adequate treatment for these pathologies [3].

Depression is a globally prevalent illness, affecting 3.8% of the population, which means that 280 million people worldwide suffer from depression [4]. Symptoms of depression are related to lack of interest and energy, loss of concentration, and feelings of guilt and worthlessness [5]. This pathology can negatively influence the affected person’s school, work, and family activities, where a worsening of the condition could lead to recurrent thoughts of death and even suicide [6]. In Spain, the estimated social costs of depressive disorders in the Spanish population were EUR 6145 million, with an average cost per patient/year of EUR 3402 [7].

In addition, research with adults shows that depression is a risk factor for cardiovascular disease [8,9]. Health indicators related to depressive symptoms in adolescence and early adulthood have begun to be identified. In this regard, it is suggested that depressive symptoms may be precursors of cardiovascular disease health risks [10].

Depression and anxiety disorders are among the most prevalent and debilitating mental health conditions (3.8–25%) [11]. There are several subtypes of anxiety disorders whose severity or impact can negatively affect an individual’s well-being and daily life, characterised by an aversive state of worry that becomes severe and persistent [12]. If left untreated, the personal and social costs of anxiety are significant, associated with frequent primary and acute care visits, reduced work productivity, and impaired social relationships [13]. Anxiety also influences cardiovascular disease, being associated with a 26% increase in the risk of coronary heart disease and a 48% increase in the risk of cardiac death [14].

Tobacco is currently considered a global health risk factor. Tobacco use is the most important preventable cause of death in the world, due to its prevalence and its association with numerous causes of death [15]. Cigarette smoke contains dozens of complex chemical substances that can give rise to relevant pathologies, such as 17 types of cancers (lung cancer being the most prominent), asthma, COPD, mental illness, and pulmonary fibrosis, as well as increasing the risk of coronary artery disease and atherosclerosis [16,17]. In Spain, 28.2% of men and 20.8% of women over the age of 16 were smokers in 2017 and, according to data from the National Statistics Institute (INE), diseases of the circulatory system, respiratory pathologies, and tumours were the main causes of death in the country, most of which were associated with tobacco use [18].

The link between smoking and depression is consistent, as the action of smoking is associated with stress relief in the smoker [19]. In several epidemiological studies, smoking is often comorbid with major depression [20]. Furthermore, smoking has been found to increase the risk of depression [21].

There is cross-sectional evidence that a variety of mental disorders (schizophrenia, depression, anxiety, bipolar disorder…) are related to detrimental health behaviours, such as low levels of physical activity (PA), poorer dietary and sleep patterns, and higher rates of smoking compared to healthy individuals [22]. Physical inactivity and sedentary lifestyles have been shown to have potential detrimental effects on physical and mental health, and may have a negative impact on cognitive processes, such as memory and attention, and are a potential predictor of disorders, such as depression [23]. In line with this fact, according to the World Health Organization (WHO) in 2020, regular PA reduces depressive and anxiety symptoms [24,25].

More specifically, people with established depression tend to have low levels of moderate to vigorous intensity PA and are less likely than healthy people to meet the recommended PA guidelines [26,27]. PA and exercise are recommended as possible treatments for depression and are introduced in guidelines as a complementary approach to other treatments for mild to moderate severity of illness [28].

In relation to PA and anxiety, several studies in the general population show that people who engage in more PA have a lower risk of anxiety diagnosis and less frequent and less severe anxiety symptoms [29,30,31,32,33,34]. Physical inactivity has also been identified as a risk factor for the development of anxiety itself [35].

Therefore, the present study aims to (1) explore the prevalence of depression and anxiety in relation to sex and to the condition of smoker and ex-smoker, (2) study the possible differences in the proportions of prevalence of depression and anxiety according to the frequency (sub study 1) and level (sub study 2) of PA performed, as well as (3) calculate the risks of probability of presenting depression or anxiety in the population according to the frequency (sub study 1) and level (sub study 2) of PA performed.

According to these objectives, it is hypothesized that: (1) there will be dependency relationships between the levels of prevalence of depression and anxiety and sex in the Spanish population of smokers and ex-smokers; (2) differences in prevalence proportions will be found according to frequency (sub-study 1) and level (sub-study 2); and (3) there will be a greater risk of probability of presenting depression or anxiety in the population that performs PA less frequently.

## 2. Materials and Methods

### 2.1. Design and Procedures

#### 2.1.1. European Health Survey of Spain 2014 (EESE2014) and 2020 (EESE2020)

This is a cross-sectional study based on data extracted from the European Health Survey of Spain 2014 (EESE2014) and 2020 (EESE2020). The EESE is a survey conducted every five years with the general objective of providing information on the health status of residents in Spain. It uses a common European questionnaire [36,37] with harmonized and comparable answers at a European level. The purpose is based on planning and evaluating health-related actions. It is carried out by the National Statistics Institute (INE), the Spanish part of the European Interview Survey (EHIS), coordinated by Eurostat, and regulated by Regulation (EC) 1338/2008 and Commission Regulation 141/2013. In Spain, the questionnaires were adapted by the INE and the Ministry of Health, Consumer Affairs and Social Welfare (MSCBS). Both surveys were conducted by interviewers trained and accredited by the INE. Face-to-face interviews were conducted between January 2014 and February 2015 (EESE2014) and on 15 July 2019 and 24 July 2020 (EESE2020).

#### 2.1.2. Spanish National Health Survey 2017 (ENSE2017)

Data extracted from the 2017 Spanish National Health Survey (ENSE2017) [38]. The ENSE is a survey conducted in Spain every five years by the MSCBS with the collaboration of the INE. This survey collects health information on the resident population of Spain. The purpose of the survey is to find out health indicators for citizens in order to help in the planning and evaluation of health-related actions. The surveys were conducted by staff trained and accredited by the MSCBS between October 2016 and October 2017. The sample was selected using a three-phase stratified random sampling system. Selected participants were informed of their inclusion in the survey and the objectives of the survey and agreed to participate prior to the interviews [38].

### 2.2. Participants

#### 2.2.1. European Health Survey of Spain 2014 (EESE2014) and 2020 (EESE2020)

Based on the EESE2014 and EESE2020 microdata published by INE and MSCBS, an initial sample of 44,914 participants was obtained (the list of participants is shown in Figure 1): EESE2014 (22,842) and EESE2020 (22,072); residents in Spain over 15 years of age. People aged 70 years and older could not be analysed because the design of the ENSE2017 did not include questions related to PA, corresponding to the IPAQ Short, for people aged 70 years and older. For this reason, people in this age range were also not included in the EESE 2014 and EESE2020. For the EESE, participants were selected based on an automated randomized system described in the methodology of both surveys [36,37]. The inclusion criteria for this research were: being under 70 years of age at the time of the survey, being a regular or ex-regular smoker, submitting data on their depression-anxiety status, and submitting data on the frequency of PA performed. We excluded 27,577 participants (13,318 from EESE2014 and 14,259 from EESE2020) for being older than 70 years (4825 and 5456), not submitting data on smoking status (24 and 25), being a non-smoker (7988 and 8353), occasional smoker (466 and 415), no data on depression status (7 and 6), no data on anxiety status (1 and 3), and no data on frequency of PA performed (7 and 1).

Finally, the sample was 17,337 participants: 9524 from EESE2014 (4763 smokers and 4761 ex-smokers; 5255 men and 4269 women; median age 47 (range: 19)) and 7813 from ESE2020 (3887 smokers and 3926 ex-smokers; 4308 men and 3505 women; median age 51 (range: 18)).

#### 2.2.2. Spanish National Health Survey 2017 (ENSE2017)

This research had an initial sample of 23,089 participants from the ENSE2017, whose data were included in the microdata published by the MSCBS. The participants, who were residents of Spain, ranged in age from 15 to 103 years (the list of participants is shown in Figure 2). Inclusion criteria were: being under 70 years of age at the time of the survey (as they were not questioned about their PA in the ENSE2017), being a smoker or having been a regular smoker, and presenting all the data on PA performed. Therefore, participants were excluded: older than 70 years (5312), non-smokers (8198), occasional smokers (466), no data on smoking status (11), and no data on PA variables (23). For analyses that included depression and chronic anxiety status, participants who did not submit data on the variable including depression (6) and chronic anxiety (9) were excluded. These participants were included in the remaining analyses. The final sample was 9079 participants: 4590 smokers (median age 46 years and interquartile range 18 years) and 4489 ex-smokers (median age 52 years and interquartile range 18 years); 4989 men and 4090 women.

### 2.3. Variables

#### 2.3.1. European Health Survey of Spain 2014 (EESE2014) and 2020 (EESE2020)

Smoking status: data extracted from item 121 of both surveys “Can you tell me if you smoke? Ignore e-cigarettes or other similar electronic devices”. Possible answers: “Yes, I smoke daily”; “Yes, I smoke, but not daily”; “I do not currently smoke, but I have smoked once”; “I do not smoke, nor have I ever smoked regularly”; “Don’t know”; or “No answer” (NS/NC). For this research we considered: smokers (those who answered “Yes, I smoke daily”) and ex-smokers (those who answered “I do not currently smoke, but I have smoked before”).

Frequency of physical activity (FPA): data extracted from item 112 of both surveys “Which of these possibilities best describes the frequency with which you do some PA in your free time? Based on the answers given, the following frequency was considered: Inactive (“I don’t exercise. I spend my free time almost entirely sedentary: reading, watching TV, going to the cinema, etc.”); Occasional (“I do some PA or sport occasionally (walking or cycling, gentle gymnastics, recreational activities involving light exertion, etc.”); Active (“I do PA several times a month: sports, gymnastics, jogging, swimming, cycling, team games, etc.”); and Very active (“I do sport or physical training several times a week”). Participants who answered “NS/NC” were excluded.

Depression status: Data were obtained from item 25a.21 of both surveys “I am going to read you a list of illnesses or health problems. Do you suffer or have you ever suffered from depression?”. Possible answers: “Yes”, “No”, or “NS/NC”.

Anxiety status: Data were obtained from item 25a.21 of both surveys “I am going to read you a list of illnesses or health problems. Do you suffer or have you ever suffered from chronic anxiety?”. Possible answers: “Yes”, “No”, or “NS/NC”.

#### 2.3.2. Spanish National Health Survey 2017 (ENSE2017)

Smoking status: data extracted from item 121 of the ENSE2017 “Can you tell me if you smoke?”. Possible answers: “Yes, I smoke daily”; “Yes, I smoke, but not daily”; “I do not currently smoke, but I have smoked once”; “I do not smoke, nor have I ever smoked regularly”; “I don’t know”; or “No answer” (NS/NC). For this research we considered: smokers (those who answered “Yes, I smoke daily”) and ex-smokers (those who answered “I do not currently smoke, but I have smoked before”).

Depression status: data were obtained from item 25a.21 of the ENSE2017 “I am going to read you a list of illnesses or health problems. Do you suffer or have you ever suffered from depression?”. Possible answers: “Yes”, “No”, or “NS/NC”.

Anxiety status: data were obtained from item 25a.21 of the ENSE2017 “I am going to read you a list of illnesses or health problems. Do you suffer or have you ever suffered from chronic anxiety?” Possible answers: “Yes”, “No”, or “NS/NC”.

Physical activity level (PAL): PA levels were: inactive, walker, active, and very active. These levels grouped participants according to self-reported PA in the questions corresponding to the IPAQ (International Physical Activity Questionnaire) Short, Spanish version, items 113–117 of the ENSE2017. With the answers given to items 113 (“First of all, think about the intense activities you did in the last 7 days. Intense activities are those that require great physical exertion and make you breathe much harder than normal, such as heavy lifting, digging, aerobic exercise or fast cycling. Think only of those that you did for at least 10 min at a time. During the last 7 days, on how many days did you engage in vigorous physical activity?”), 114 (“On one of these days, how much time in total did you spend in intense physical activity?), 115 (“Now please think of all those moderate activities you did in the last 7 days. Moderate activities are those that require moderate physical exertion that makes you breathe a little harder than normal, such as carrying light weights, cycling at regular speed or playing tennis doubles. Think only of those that you did for at least 10 min at a time. During the last 7 days, on how many days did you do moderate physical activity? Please do not include walking.”) and 116 (“On one of these days, how much time in total did you spend in moderate physical activity?”) a Physical Activity Index (IAF, by its Spanish acronym) was created, according to the index created by Denche et al. [39]. This index was adapted from the Physical Activity Index (PAI) by Ness et al. [40], whose formula is: IAF = Intense activity score + Moderate activity score. We considered: Inactive (participants with IAF = 0, who responded, no day to item 117 “Now think about how much time you spent walking in the last 7 days. This includes walking at work, at home, to get from one place to another, or whatever you walk for sport, exercise or pleasure. Think only of those occasions when you walked for at least 10 min at a time”), Walker (participants with IAF = 0, you want to respond to item 117 by walking at least one day for more than 10 min at a time), Active (IAF between 1 and 30) and Very active (IAF higher than 30). IAF scores could range from 0 to 67.5 points.

### 2.4. Statistical Analyses

The distribution followed by the data of the variables of interest was analysed with the Kolgomorov–Smirnov test, without finding sufficient evidence to assume normality. Following the results obtained in the normality test, the data of the variables were presented through absolute and relative frequencies in the following descriptive analysis. In addition, non-parametric statistical tests were performed. The sample was characterised, showing the prevalence of depression and anxiety in the general smoking and ex-smoking population, as well as by sex. The dependence between the prevalence of depression and anxiety and sex was analysed with the Chi-square test and possible differences in proportions between sexes with the z-test for independent proportions. The same was performed to assess the dependence between the prevalence of depression and anxiety and the self-reported PA frequency. The odds of anxiety and depression were calculated for smokers and ex-smokers, according to the frequency of PA, taking the very active group as a reference. A correlation study was carried out using Spearman’s rho between the variables of interest. The correlation coefficients were interpreted according to Barrera’s proposal [41]. IBM SPSS Statistics v.25 for Windows (IBM Corp., Armonk, NY, USA) statistical software was used, assuming a significance level of less than 0.05. 

## 3. Results

### 3.1. European Health Survey of Spain 2014 (EESE2014) and 2020 (EESE2020)

Table A1 shows the dependency relationships between the levels of prevalence of depression and anxiety and sex in the Spanish smoking and ex-smoking populations in the years 2014 and 2020, according to the EESE, *p* < 0.001 in the Chi-square test. Significant differences were found in the proportions of people with depression and anxiety, according to sex (*p* < 0.05 in the z-test), being higher in women than in men, both in smokers and ex-smokers (Figure 3).

In Table A2, dependency relationships were found between the prevalence of depression and anxiety with the frequency of PA (*p* < 0.001 in the Chi-Square test) in both smokers and ex-smokers.

In smokers in 2014 (Figure 4), the highest differences in prevalence of depression (15.2%) and anxiety (13.3%) were found in inactive people, and the lowest in active people (5.4% and 5.0%, respectively), with these differences in proportions being statistically significant (*p* < 0.05 z-test). Among ex-smokers, the highest prevalence of depression (11.8%) and anxiety (11.3%) was also found among inactive people. The lowest prevalence of depression (6.6%) was in the very active, and the lowest prevalence of anxiety (5.7%) was in the active (*p* < 0.05 at the z-test).

In smokers in 2020 (Figure 5), the largest differences in prevalence of depression (13.2% vs. 5.9%) and anxiety (12.9% vs. 6.7%) were found in inactive versus active persons, (*p* < 0.05 at z-test). In ex-smokers, the higher prevalence of depression (12.2% vs. 4.5%) and anxiety (11.5% vs. 6.5%) was also found in inactive vs. active people (*p* < 0.05 at the z-test).

Table A3 shows the odds ratios (OR) for depression and anxiety according to the frequency of PA, taking as a reference the very active group in smokers and ex-smokers in the EESE 2014 and 2020.

In 2014, inactive people had elevated risks of depression compared to very active people, in both smokers (OR: 2.06; 95%CI: 1.45–2.94) and ex-smokers (OR: 1.88; 95%CI: 1.33–2.65). The risk of anxiety was also found to be elevated in inactive smokers (OR: 3.56, 95%CI: 2.49–5.09) and ex-smokers (OR: 1.92, 95%CI: 1.34–2.74) compared to very active smokers.

In 2020, inactive people had elevated risks of depression compared to very active people, in smokers (OR: 1.83; 95%CI: 1.26–2.68) and in ex-smokers (OR: 2.29; 95%CI: 1.59–3.30). The risk of anxiety was also found to be elevated in inactive smokers (OR: 1.98, 95%CI: 1.34–2.94) and ex-smokers (OR: 1.80, 95%CI: 1.27–2.55) compared to very active smokers.

In Table A4, in both smokers and ex-smokers in EESE 2014, medium correlations were shown between frequency of physical activity and age, and weak correlations between frequency of physical activity and prevalence of depression, prevalence of anxiety, and gender. In EESE 2020, in both smokers and ex-smokers, weak correlations were shown between physical activity frequency and the other variables.

### 3.2. Spanish National Health Survey 2017 (ENSE2017)

Table A5 shows the dependency relationships between the levels of prevalence of depression and anxiety and sex in the Spanish smoking and ex-smoking populations in 2017 according to the ENSE, *p* < 0.001 in the Chi-square test. Significant differences were found in the proportions of people with depression and anxiety, according to sex (*p* < 0.05 in the z-test), being higher in women than in men, both in smokers and ex-smokers.

In Table A6, dependence relationships were found between the prevalence of depression and anxiety with the PA level (*p* < 0.001 in the Chi-Square test), both in smokers and ex-smokers.

In smokers (Figure 6), the highest differences in prevalence of depression (16.3%) and anxiety (16.6%) were found in inactive people, and the lowest in very active people (7.0% and 7.4%, respectively), these differences in proportions being statistically significant (*p* < 0.05 at the z-test). The same was found in ex-smokers, with a higher prevalence of depression (16.7% vs. 3.9%) and anxiety (15.5% vs. 5.7%) in inactive versus very active smokers (*p* < 0.05 at z-test).

Table A7 shows the odds ratios (OR) for depression and anxiety according to the PA level, taking as a reference the very active group in smokers and ex-smokers in the ENSE 2017.

Inactive people had elevated risks of depression compared to very active people, in smokers (OR: 2.60, 95%CI: 1.74–3.88) and ex-smokers (OR: 4.91, 95%CI: 3.04–7.94). The risk of anxiety was also found to be elevated in inactive smokers (OR: 2.49, 95%CI: 1.68–3.68) and ex-smokers (OR: 3.03, 95%CI: 1.98–4.62) compared to very active smokers.

In Table A8, for smokers, mean correlations between physical activity level and age, and weak correlations between physical activity level and prevalence of depression, prevalence of anxiety, and gender are revealed. For ex-smokers, mean correlations were shown between physical activity level and prevalence of depression and age, and weak correlations between physical activity level and prevalence of anxiety and gender.

## 4. Discussion

Among the main findings on the levels of prevalence of depression and anxiety in the Spanish smoking and ex-smoking population aged 15–70 years, dependence relationships between levels of prevalence and sex of the participants were shown. Thus, female smokers and ex-smokers had a significantly higher proportion of sufferers of depression and anxiety than men. Dependence relationships were also found between the prevalence of depression and anxiety with the frequency of PA (EESE 2014 and 2020) and the PA level (ENSE 2017) in smokers and ex-smokers. In general, levels of prevalence were higher in inactive versus very active or active people, whose differences in proportions were statistically significant. According to the data collected, being a smoker or ex-smoker and, in addition, being “inactive” from the point of view of PA could imply a higher risk of probability of suffering from depression and anxiety compared to “very active” people. Furthermore, in the EESE 2020 and in the ENSE 2017, the risk of depression was higher in ex-smokers than in smokers, with the highest risk in ex-smokers (OR: 4.91; 95%CI: 3.04–7.94) in the ENSE 2017. As for anxiety in the ENSE 2017, the risk of anxiety was also higher in ex-smokers than in smokers.

In both sub-study 1 and sub-study 2, in smokers and ex-smokers, a higher prevalence of depression (9.4–15.7% vs. 6.8–9.1%) and anxiety (11.4–16.3% vs. 5.8–7.9%) was found in women compared to men. Another study in the general population also found a higher prevalence of depression in women (12.5%) than in men (6.2%), in this case twice as high [42].

A slightly higher prevalence of depression and anxiety in smokers than in ex-smokers was observed in all three surveys in the present study. Moreover, these results in both groups (depression 12% to 8%; anxiety 11.8% to 8.3%) can be considered slightly higher when compared to general population data in Spain from other studies. For example, in the study by Vieta et al., the mean annual prevalence of depression estimated during 2015–2017 was 4.73% [7], and with the EESE 2020 where chronic anxiety was 5.8% and depression was 5.3% [43]. Furthermore, if we compare the levels of prevalence of depression and anxiety in smokers and ex-smokers with other populations, such as older Spanish adults (people over 65 years of age), we observe that the overall prevalence of anxiety (11%) is similar to those in the present research [44]. This may be due to the fact that older adults may interpret or experience affective terms differently and have a higher health concern than younger adults [45]. In other pathologies, such as type 2 diabetes mellitus, the prevalence of depression was higher (20.03 %), these higher values are possibly associated with poor adherence to medication for blood glucose, blood pressure, and cholesterol, among other factors [46,47]. In inflammatory bowel disease, a non-curable condition, the levels of prevalence were higher (anxiety 32.1% and depression was 25.2%) [48]. Thus, in smokers and ex-smokers, having an associated pathology may even increase the risk of depression or anxiety.

This is supported by Luger et al., who highlighted in their research that there were longitudinal associations between smoking and subsequent risk of depression across four meta-analyses of 19 studies for a total of 79,729 participants. In these, smoking significantly increased the risk of depression (smokers compared to never smokers), measured as diagnosed depressive disorders or cynically significant depressive symptoms on validated scales (OR = 1.62; 95 % CI: 1.1–2.4). A similar situation was found between smokers and ex-smokers, where smokers were more likely to be depressed than ex-smokers (OR = 1.76, 95 % CI: 1.48–2.09) [49].

In relation to PA and the prevalence of depression and anxiety in both smokers and ex-smokers, the highest prevalence was found in inactive people, i.e., people who do not exercise and spend their free time doing sedentary activities. The highest prevalence of depression in all three surveys was found in inactive ex-smokers, with 16.7%, which is remarkable since the usual tendency was for smokers to have higher levels of prevalence than ex-smokers. The highest prevalence of anxiety was in inactive smokers, with 16.6%. In relation to other research, Galán-Arroyo et al. highlighted that, in the general population, the prevalence of depression and anxiety were associated with the frequency of PA. The highest levels of prevalence of depression were also found in the population that “never” performs PA (13.4%), however, these data were slightly lower than in the present study, as the main difference lay in the fact that they studied the general population, as opposed to the condition of being a smoker or ex-smoker in the present study [42]. This study by Galán-Arroyo et al. also shows that there is a lower prevalence of depression in the “active” population, which is noteworthy as they have lower prevalence data than the “very active” population, as in our study. This could be due to the fact that excessive PA could lead to overtraining and trigger psychological symptoms that mimic depression [50].

In this line, Schuch et al., in their meta-analysis of 36 prospective comparisons, observed that higher levels of PA significantly decreased the subsequent risk of depressive events during a mean follow-up time of 7 years (comparison between higher levels of PA with lower levels) (OR = 0.837, 95 % CI: 0.794 to 0.883) [51], in their meta-analysis in 11 cohorts with a total of 69,037 participants, found that higher levels of PA significantly reduced incident anxiety (OR = 0.748, 95 % CI 0.629–0.889) over a follow-up period of 3.5 years [52]. These data are consistent with those of the present study and the frequency of PA or the PA level where, compared to “very active” smokers and ex-smokers, “inactive” smokers had elevated risks of depression and anxiety.

As noted, physical inactivity may increase the likelihood of depression and anxiety, however, the possibility of reverse causality, i.e., that depression and anxiety are precursors of physical inactivity itself, should be considered, which was not taken into account in the present study. Hartanto et al. show an inverse causal relationship between social networks and depressive symptoms. Cross-sectional and correlational trend analyses appear to show increases in rates of depression along with increased participation in social networks. However, depressive symptoms may precede increased use of social networks, using them as an escape or for self-esteem reaffirmation through social validation [53]. In relation to our research, depression and anxiety could precede physical inactivity, and symptoms of these mental illnesses could influence a possible demotivation towards PA, i.e., suffering from depression and anxiety could reduce the levels or frequency of PA. It would be interesting to study this assumption in future research as there is not much scientific evidence in this respect.

Therefore, increasing frequency of PA or PA level in smokers and ex-smokers could lead to a reduction in the risk of depression and anxiety. To this end, the possible practical implications of this study could be used by public administrations, as well as by national and regional health services, favouring the reduction in the prevalence of depression and anxiety in all age ranges through awareness campaigns focused on health promotion, training and educating people on the importance of PA to reduce the symptomatology of these mental illnesses.

Promoting an increase in frequency of PA or PA level in smokers and inactive ex-smokers would facilitate a decrease in the OR of developing depression or anxiety. Therefore, from the public sphere, it would be advisable to invest in health education throughout the life cycle, introducing sports science professionals in health services, who could prescribe physical exercise adapted to the needs of each patient. In addition, in the private (or public) sector, the inclusion by public organizations and/or companies of PA programmes based on active breaks could increase PA time in the workplace and/or in education. Therefore, these measures could have an impact on reducing the prevalence of these mental disorders for people who do not do PA due to lack of free time. It could even reduce health and social costs in the long term.

This study has a series of limitations, among them we can highlight the lack of differentiation between sexes in the study of the association of PA level with the risk of suffering from depression and anxiety. Therefore, a possible interesting future line of research would be to explore how PA variables affect each sex when differences in the prevalence of depression and anxiety between men and women are detected.

As this was a cross-sectional study, it was not possible to establish cause–effect relationships. It would be advisable to further explore the findings through other research to establish causal relationships. The study is based on self-reported PA data, and PA may be overestimated when measured subjectively. In subsequent studies, it would be innovative to include objective PA data from the participants, using appropriate methods and instruments. Furthermore, the type of training undertaken by each individual was not specified in the surveys, which may have a bearing on the risk of depression and anxiety

The study was based on self-reported depression and anxiety, and not on diagnostic tests, so the results could be affected. As these are national and European surveys, there are many factors and aspects that are collected within them, including for our research the questions and sections that were of most interest to us for the elaboration of this study. In future studies, it would be interesting to use other more complete or objective methods to assess the suffering from these mental illnesses, or even to include participants suffering from depression or anxiety from the outset, whose diagnosis has been made by a specialist.

Only male and female sex was considered, as non-binary sex was not taken into account. Whether participants were taking antidepressants was not taken into account. Other variables that could have an impact on depression, such as socio-demographic, socio-cultural and socio-economic biases of the participants, were also not included [54]. In future studies, in order to analyse the influence of physical activity on the prevalence rates of depression and anxiety, it would be interesting to divide the sample according to the age ranges and social status of the participants, as other studies have shown that it could have an influence on the reduction or increase in these rates [55].

Finally, another interesting future line of research would be to study the prevalence of depression and anxiety and their association with the level of PA in different pathologies, especially those that could be associated with smoking, such as COPD or other cardiorespiratory diseases.

## 5. Conclusions

It can be concluded that physically inactive smokers and ex-smokers in Spain had higher levels of prevalence of depression and anxiety than active and very active people, so that physical inactivity could increase the risk of suffering from these mental disorders in this population. This could affect women more than men, as a higher prevalence of depression and anxiety was detected in women than in men, both smokers and ex-smokers.

## Figures and Tables

**Figure 1 healthcare-10-01403-f001:**
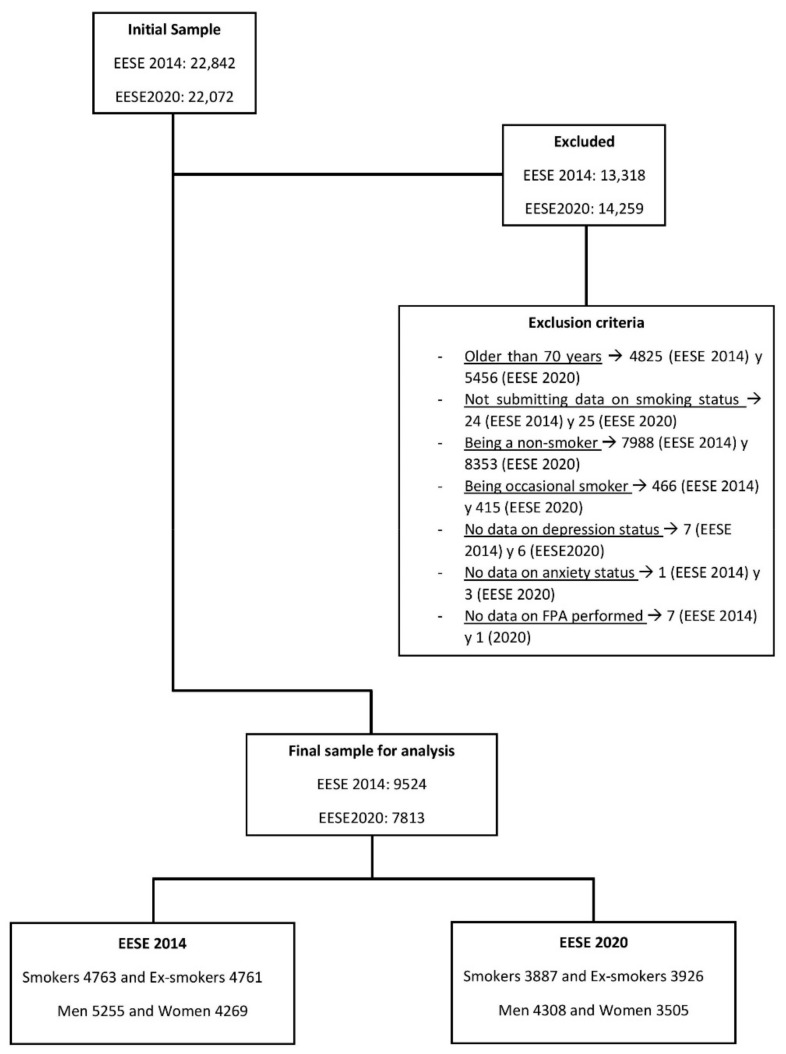
Sample exclusion criteria (sub-study 1).

**Figure 2 healthcare-10-01403-f002:**
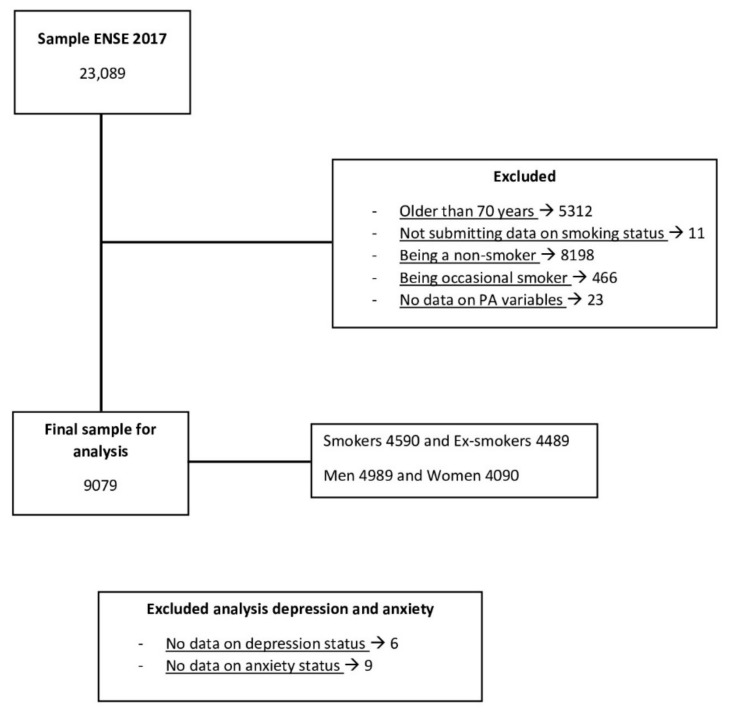
Sample exclusion criteria (sub-study 2).

**Figure 3 healthcare-10-01403-f003:**
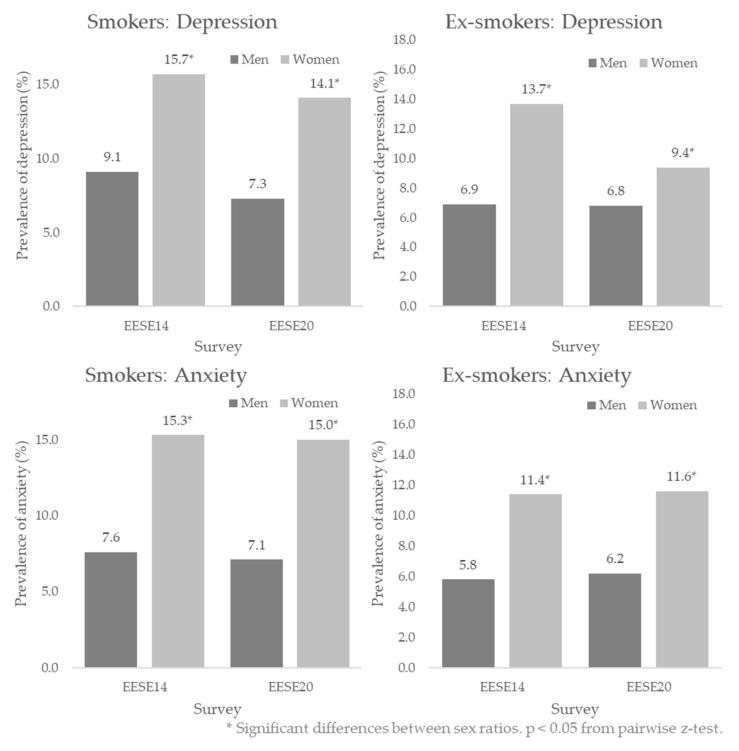
Prevalence of depression and anxiety in Spanish smokers and ex-smokers, and comparison by sex: EESE2014 and EESE2020.

**Figure 4 healthcare-10-01403-f004:**
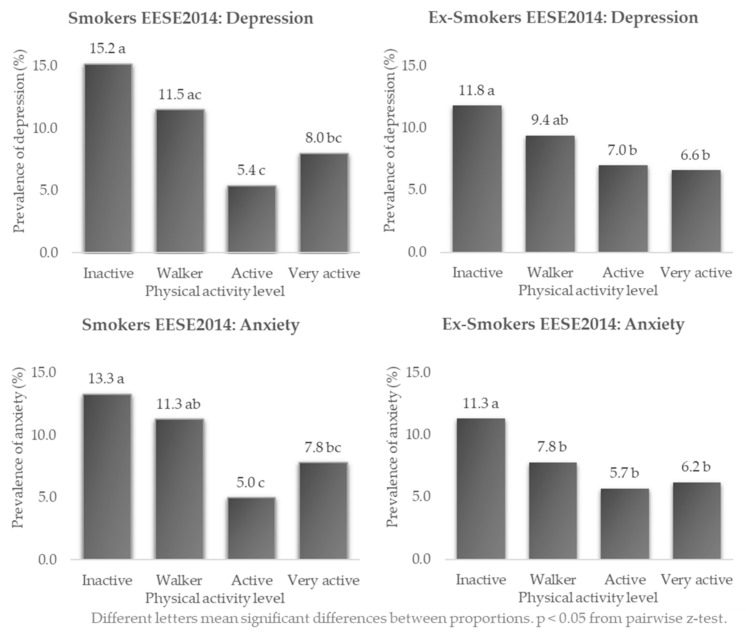
Prevalence of depression and anxiety in Spanish smokers and ex-smokers according to their frequency of physical activity: EESE2014.

**Figure 5 healthcare-10-01403-f005:**
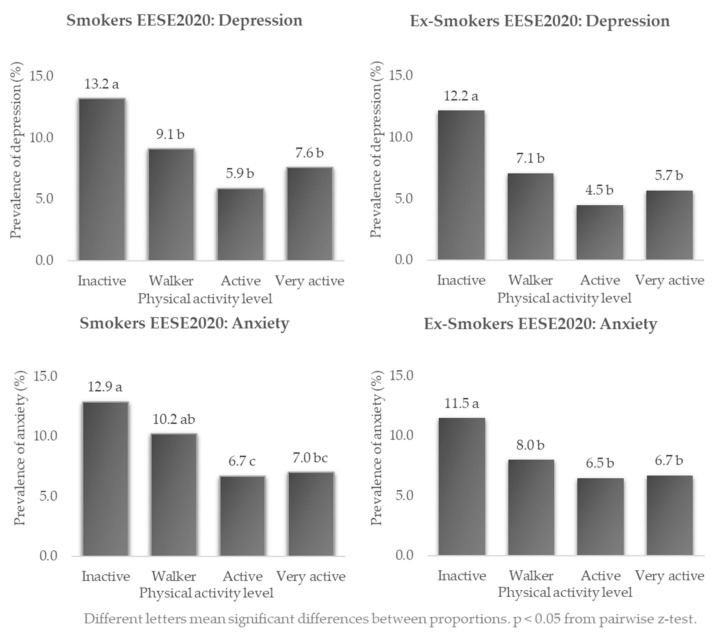
Prevalence of depression and anxiety in Spanish smokers and ex-smokers according to their frequency of physical activity: EESE2020.

**Figure 6 healthcare-10-01403-f006:**
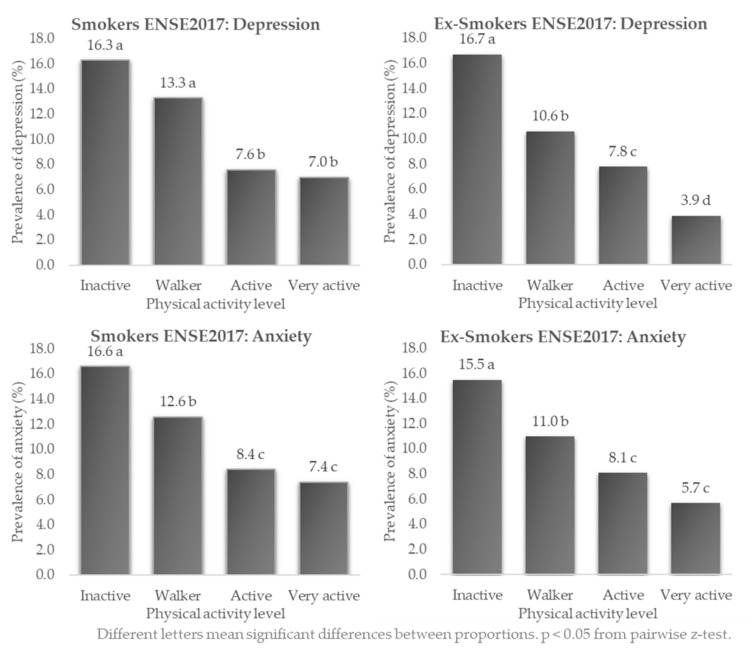
Prevalence of depression and anxiety in Spanish smokers and ex-smokers according to physical activity level. ENSE2017.

## Data Availability

Datasets will be available under reasonable request.

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
