# Peer review of "A Cross-Sectional Study on the Associations between Physical Activity Level, Depression, and Anxiety in Smokers and Ex-Smokers"

_healthcare, 2022, doi:10.3390/healthcare10081403_

Round 1

Reviewer 1 Report

GENERAL

Well written manuscript which focuses on an important research questions and significantly contributes to public health research area.

METHODS:

EESE2014 and EESE2020 and ENSE2017 : Why is the age threshold below 70 years?

Statistical analyses: Did you calculate a logistic regression for the odds ratios? If so, please mention this in the text.

FORMALITIES

According to APA you write p-values without a zero before the comma because the p-value always lies between zero and one. Please correct.

p. 3, l.113 & l.127: It does not look nice to have a paragraph of a single sentence. Please combine the sentences with the paragraph bellow.

I would not write „substudy“ in the Methods section as this implies a main study and then a further substudy which is not your case. A better word could be „secondary data analysis“. I would also not oder the text in sub-chapters for each „substudy“, but suggest to name the chapters after the name of each survey.

p. 7, l.244 It does not look nice to have 2 paragraphs here. I understand that it is important to mention the paragraph l.245 – l.251, but it looks like you forgot to write about it and then just added it anywhere in the manuscript. Please combine these to paragraphs into one paragraph.

Tables are well designed and include sufficient description of the data they are presenting.

Author Response

Dear Reviewer:

We appreciate your words about our work and all your comments in order to improve our manuscript.

GENERAL

Well written manuscript which focuses on an important research questions and significantly contributes to public health research area.

  • Author’s response: Thank you for your review of our manuscript. We have carefully considered your comments and believe that the quality of the paper has improved after incorporating your suggestions. Below are our responses to your suggestions:

METHODS:

EESE2014 and EESE2020 and ENSE2017 : Why is the age threshold below 70 years?

  • Author’s response: In the design of the Spanish National Health Survey 2017, questions related to physical activity, corresponding to the IPAQ Short, were not included for those aged 70 years and older. Therefore, we were not able to perform the analyses with those over 70 years of age and this was the reason why we decided not to include those over 70 years of age from the other surveys. We have included an explanation of your suggestion in the participants' section. Thank you very much for your comment.

Statistical analyses: Did you calculate a logistic regression for the odds ratios? If so, please mention this in the text.

  • Author’s response: Odds ratios were not calculated using a logistic regression model. Let's hope we have answered your question

 FORMALITIES

According to APA you write p-values without a zero before the comma because the p-value always lies between zero and one. Please correct.

  • Author’s response: Thank you for your appreciation, this has been corrected.
  1. 3, l.113 & l.127: It does not look nice to have a paragraph of a single sentence. Please combine the sentences with the paragraph bellow.
  • Author’s response: The paragraphs have been combined, thank you for your comment.

 I would not write „substudy“ in the Methods section as this implies a main study and then a further substudy which is not your case. A better word could be „secondary data analysis“. I would also not oder the text in sub-chapters for each „substudy“, but suggest to name the chapters after the name of each survey.

  • Author’s response: We have renamed the sections, thank you for your contribution.
  1. 7, l.244 It does not look nice to have 2 paragraphs here. I understand that it is important to mention the paragraph l.245 – l.251, but it looks like you forgot to write about it and then just added it anywhere in the manuscript. Please combine these to paragraphs into one paragraph.
  • Author’s response: It was an edition error, the paragraph was in the previous paragraph. We have corrected it. Thank you for your comment.

Tables are well designed and include sufficient description of the data they are presenting.

  • Author’s response: Thank you for your time, your comments and for contributing to the improvement of our manuscript.

Reviewer 2 Report

This is an interesting study examining the associations between physical activity, depression and anxiety in smokers and ex-smokers. I appreciate the large sample of the study. I can see the contribution of this study to the literature. However, I have several comments that require the authors' attention:

1. It will be important for the authors to provide more information on how the sample was recruited. There is a mention that three-phase stratified random sampling system was used. More details will be helpful for readers to evaluate the representativeness of this study.

2. It seems that listwise deletion is used to handle missing data. There is a need to justify the decision to use listwise deletion. There are methods other than listwise deletion that are less bias such as multiple imputation. Perhaps, a bit more elaboration will be helpful

Relevant paper: Newman, D. A. (2014). Missing data: Five practical guidelines. Organizational Research Methods, 17(4), 372-411.

3. In the limitation section, there is a need for the authors to discuss the validity of the depression and anxiety measures. The measures seem too simplistic to measure the complexity of depression and anxiety.

4. Another important point to discuss in the Discussion section is the possibility of reverse causation which is a common limitation in the literature. While the authors suggested that inactivity could lead to depression and anxiety, it is equally plausible that depression and anxiety are the antecedent of inactivity. This is a widespread issue in a depression research. This problem related to reverse causation should be discussed in details to provide a more balanced interpretation of the result. This is a critical point to discuss. Please see the following paper for a relevant discussion related to reverse causation: Quek, F. Y., Tng, G. Y., & Yong, J. C. (2021). Does social media use increase depressive symptoms? A reverse causation perspective. Frontiers in Psychiatry, 12, 335.

5. The use of abbreviation may disrupt the reading flow. I would suggest the authors to avoid abbreviation such as PAL and FPA

6. There is a need for the authors to control for age and social status in their analyses to ensure the robustness of their results. Miech, R. A., & Shanahan, M. J. (2000). Socioeconomic status and depression over the life course. Journal of Health and Social Behavior, 162-176.

7. It will be useful for the authors to report zero-order correlations of all main variables in the study.

8. More information regarding the participants' characteristics (e.g., education, income, marital status) will be useful in the method section.

Author Response

Dear reviewer:

Thank you very much for your kind words about our manuscript, as well as for all your efforts to contribute to improving the presentation of our research in it.

This is an interesting study examining the associations between physical activity, depression and anxiety in smokers and ex-smokers. I appreciate the large sample of the study. I can see the contribution of this study to the literature. However, I have several comments that require the authors' attention:

  1. It will be important for the authors to provide more information on how the sample was recruited. There is a mention that three-phase stratified random sampling system was used. More details will be helpful for readers to evaluate the representativeness of this study.
  • Author’s response: In the section "Participants" we had included the references to the survey methodologies, where there is an extensive description of the sampling system (line 142). Again, we have added the reference to the original document in line 133 to prevent confusion and so that the reader can refer to the document where the entire sampling process is explained in detail. If you think that further specification is needed, please let us know.
  1. It seems that listwise deletion is used to handle missing data. There is a need to justify the decision to use listwise deletion. There are methods other than listwise deletion that are less bias such as multiple imputation. Perhaps, a bit more elaboration will be helpful

Relevant paper: Newman, D. A. (2014). Missing data: Five practical guidelines. Organizational Research Methods17(4), 372-411.

  • Author’s response: Thank you very much for this contribution, we understand that the proposed method of multiple imputation can be applied. However, we have chosen to use the current analysis. Multiple imputation would imply a more tedious analysis and even generate compression problems for readers as it is composed of a multitude of models (for depression, for anxiety, for the different surveys and separating according to smokers and ex-smokers) and variables, generating tables that are too large. However, if you think it is key to include these models, we will do so without any problem.
  1. In the limitation section, there is a need for the authors to discuss the validity of the depression and anxiety measures. The measures seem too simplistic to measure the complexity of depression and anxiety.
  • Author’s response: This study was based on participants' self-reported depression and anxiety. We agree that depressive and anxiety states are complex conditions and have expanded our limitations section in response to your comment. Thank you for your input and let us know if what you suggested corresponds to what you have recently added.
  1. Another important point to discuss in the Discussion section is the possibility of reverse causation which is a common limitation in the literature. While the authors suggested that inactivity could lead to depression and anxiety, it is equally plausible that depression and anxiety are the antecedent of inactivity. This is a widespread issue in a depression research. This problem related to reverse causation should be discussed in details to provide a more balanced interpretation of the result. This is a critical point to discuss. Please see the following paper for a relevant discussion related to reverse causation: Quek, F. Y., Tng, G. Y., & Yong, J. C. (2021). Does social media use increase depressive symptoms? A reverse causation perspective. Frontiers in Psychiatry12, 335.
  • Author’s response: Thank you very much for your appreciation and recommendations, it is a quality comment that has allowed us to improve our discussion. We are waiting for a response to what we have added in this respect.
  1. The use of abbreviation may disrupt the reading flow. I would suggest the authors to avoid abbreviation such as PAL and FPA
  • Author’s response: Following your recommendations, we have removed the abbreviations. Thank you very much.
  1. There is a need for the authors to control for age and social status in their analyses to ensure the robustness of their results. Miech, R. A., & Shanahan, M. J. (2000). Socioeconomic status and depression over the life course. Journal of Health and Social Behavior, 162-176.
  • Author’s response: In the section on limitations we have already written something related to this in the past. We understand what you are suggesting, however, incorporating these variables would mean restructuring our research. We have expanded the limitations section by commenting on this aspect and have proposed your suggestion for future research. Thank you for your comment
  1. It will be useful for the authors to report zero-order correlations of all main variables in the study.
  • Author’s response: Thank you for your input. Following your suggestion, we have included in the results section a correlation study using Spearman's rho between the variables of interest. We have also prepared the corresponding tables in the appendix.
  1. More information regarding the participants' characteristics (e.g., education, income, marital status) will be useful in the method section.
  • Author’s response: Commented on suggestion 6. Thank you for your input and we look forward to your response on this issue.

Round 2

Reviewer 2 Report

The authors have addressed all my comments well. I appreciate all their efforts.